# Copper Catalyst-Supported Modified Magnetic Chitosan for the Synthesis of Novel 2-Arylthio-2,3-dihydroquinazolin-4(1*H*)-one Derivatives via Chan–Lam Coupling

Nastaran Ghasemi [1], Ali Yavari [1], Saeed Bahadorikhalili [2], Ali Moazzam [1], Samanehsadat Hosseini [1], Bagher Larijani [1], Aida Iraji [3], Shahram Moradi [3] and Mohammad Mahdavi [1,*]

[1]   Endocrinology and Metabolism Research Centre, Endocrinology and Metabolism Clinical Sciences Institute, Tehran University of Medical Sciences, Tehran 1416634793, Iran
[2]   Department of Electronic Engineering, Universitat Rovira i Virgili, 43007 Tarragona, Spain
[3]   Stem Cells Technology Research Center, Shiraz University of Medical Sciences, Shiraz 7134845794, Iran
*   Correspondence: momahdavi@sina.tums.ac.ir

**Abstract:** In this paper, magnetic chitosan is used as a support for the immobilization of copper catalyst (Cu@MChit). The fabricated catalyst is successfully synthesized and characterized by several techniques. The activity of Cu@MChit catalyst is evaluated in the synthesis of novel derivatives of 3-alkyl-2-arylthio-2,3-dihydroquinazolin-4(1*H*)-ones. The products are synthesized in three simple steps via Chan–Lam coupling reaction. The synthetic route is based on the reaction of isatoic anhydride and an amine, followed by the reaction with carbon disulfide. Cu@MChit-catalyzed reaction of the obtained intermediate with phenylboronic acid leads to the desired products. The scope of the reaction is confirmed by using various amine and phenylboronic acid derivatives and the products are obtained in high isolated yields.

**Keywords:** magnetic chitosan; 2-arylthio-2,3-dihydroquinazolin-4(1*H*)-one; Chan–Lam coupling; quinazolinone; copper catalyst; chitosan; isatoic anhydride

## 1. Introduction

Quinazolinone and its derivatives, as a class of heterocycle nitrogen-containing heterocycles, are of high significance due to their wide biological activities [1,2]. Various biological activities of quinazolinones, including anticancer [3], anti-HIV [4], tyrosinase inhibiting [5], and α-Glucosidase and α-amylase inhibiting activities [6] have been confirmed. Albaconazole [7], raltitrexed [8], and methaqualone [9] are only a number of quinazolinone drugs on the market (Scheme 1).

**Scheme 1.** The structures of quinazoline-containing drugs on the market.

The most popular method for the synthesis of quinazolinone derivatives is the use of a benzene derivative bearing an amine and a carboxylic acid or cyanide group in the *ortho* positions [10,11]. Another interesting method for the synthesis of this valuable class of heterocycles is the reaction of isatoic anhydride with an amine, followed by a cyclization reaction [12–15]. We have developed several methods for the synthesis of quinazolinone, including copper-catalyzed [16–18] and palladium-catalyzed [19,20] reactions with aldehydes. Additionally, we have developed other methods based on the reactions of phenylisothiocyanate [21], chloroform, and sulfur [22], or amine and aldehyde [23] with isatoic anhydride and amine derivatives. Following our ongoing research on developing methods for the synthesis of quinazolinone derivatives, we hereby report an efficient method for the synthesis of novel 3-alkyl-2-arylthio-2,3-dihydroquinazolin-4(1*H*)-one derivatives. The method is based on the reaction of isatoic anhydride with an amine, followed by a cyclization reaction with carbon disulfide in the presence of potassium hydroxide as a catalyst. This Chan–Lam coupling [24,25] reaction of the product with phenylboronic acid in the presence of copper chloride leads to the formation of the desired products. The synthetic route of the products is presented in Scheme 2.

**Scheme 2.** Synthesis of 3-alkyl-2-arylthio-2,3-dihydroquinazolin-4(1*H*)-one derivatives from isatoic anhydride, amines, carbon disulfide, and phenylboronic acid via Chan–Lam coupling.

## 2. Results and Discussion

In this paper, a novel catalyst is synthesized based on the immobilization of copper onto modified magnetic chitosan. The synthesis of Cu@MChit is presented in Scheme 3. For the synthesis of the Cu@MChit catalyst, chitosan was first modified by superparamagnetic iron oxide nanoparticles (SPION) to form magnetic chitosan (MChit). Separately, *N*-phenylacetamide was used for the synthesis of 2-chloroquinoline-3-carbaldehyde by reacting with POCl$_3$ in dimethylformamide (DMF). The reaction of 2-chloroquinoline-3-carbaldehyde with Na$_2$S led to the formation of 2-mercaptoquinoline-3-carbaldehyde, which reacts with 3,4-diaminobenzoic acid to give 2-(2-mercaptoquinolin-3-yl)-1*H*-benzo[*d*]imidazole-6-carboxylic acid. The carboxylic acid functionality of 2-(2-mercaptoquinolin-3-yl)-1*H*-benzo[*d*]imidazole-6-carboxylic acid was activated by EDC and NHS and reacted with MChit to fabricate the Cu@MChit catalyst.

The successful synthesis of the Cu@MChit catalyst was characterized by several characterization techniques. The microstructures of the Cu@MChit catalyst were studied by TEM microscopy. Based on the TEM image, which is presented in Figure 1a, the magnetic nanoparticles are successfully supported on the chitosan. The dots represent the SPION and the brighter areas are correlated to the chitosan. In addition, FT-IR spectroscopy confirms the synthesis of the Cu@MChit catalyst. The characteristic peaks of Fe-O in SPION and also the vibrations belonging to chitosan can be observed in Figure 1b. The thermal stability of

the catalyst was studied by TGA, which is presented in Figure 1c. Based on VSM result, which can be seen in Figure 1d, the Cu@MChit catalyst is superparamagnetic and the magnetic behavior of the iron oxide nanoparticles is preserved in the catalyst. The loading of copper onto the catalyst was measured to be 1 mmol Cu per 100 mg of Cu@MChit catalyst. The stability of the structure of the catalyst was evaluated by the leaching test. This test confirmed the stability of the catalyst and no detectable copper was observed by ICP analysis.

**Scheme 3.** Synthesis of Cu@MChit catalyst.

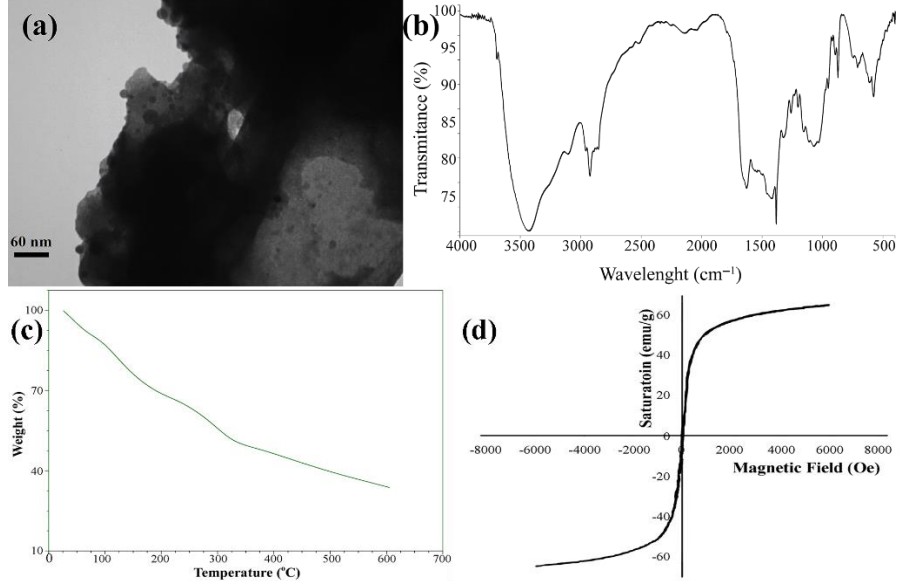

**Figure 1.** The characterization results of Cu@MChit catalyst. (**a**) TEM, (**b**) FTIR, (**c**) TGA, and (**d**) VSM results.

After the characterization of the Cu@MChit catalyst, it was used for the synthesis of novel 2-arylthio-2,3-dihydroquinazolin-4(1*H*)-one derivatives. The desired products were synthesized in three steps from commercially available chemicals. The first two steps of the reactions are well established in our lab and were performed under optimized reaction conditions. However, the conditions of the last step of the reaction were optimized for obtaining the products in the highest isolated yields.

The synthesis of 3-alkyl-2-arylthio-2,3-dihydroquinazolin-4(1*H*)-ones products are obtained in three simple reaction steps. In the first step, isatoic anhydride participated in an efficient reaction with amines for the corresponding 2-amino-*N*-alkylbenzamide derivatives (**2**). This step is a facile reaction, which is performed in water and the products are obtained in very high isolated yields. The driving force for this step is the removal of a carbon dioxide molecule from the structure of the isatoic anhydride in the synthesis of 2-amino-*N*-alkylbenzamide derivatives. The removal of gaseous carbon dioxide from the reaction mixture leads to the complete conversion of the substrates to the products.

The next reaction step involves a reaction between 2-amino-*N*-alkylbenzamides and carbon disulfide in ethanol in the presence of potassium hydroxide. This step is also efficient and the products are obtained in very good isolated yields. For optimizing the reaction conditions, the reaction of 3-allyl-2-thioxo-2,3-dihydroquinazolin-4(1*H*)-one (compound **3** with allyl group for R) and phenyl boronic acid was selected as the model reaction, and performed in various solvents, bases, and catalysts for the synthesis of 3-alkyl-2-arylthio-2,3-dihydroquinazolin-4(1*H*)-one derivatives (**4**).

The optimization results are presented in Table 1. According to the results, the best solvent for the reaction is methanol and the highest isolated yield of the product is observed in the presence of the Cu@MChit as the catalyst. In addition, the use of triethylamine base showed to be the most efficient in the mentioned synthesis. It should be noted that the desired products are also obtained in other solvents, including chloroform. However, the yield of the product showed to be very poor in chloroform.

**Table 1.** Optimization conditions of the reaction [a].

| Entry | Solvent | Base | Catalyst/Amount (mg) | Yield % [b] |
|---|---|---|---|---|
| 1 | DMF | Et$_3$N | Cu@MChit/20 | No reaction |
| 2 | CHCl$_3$ | Et$_3$N | Cu@MChit/20 | 74 |
| 3 | MeOH | Et$_3$N | Cu@MChit/20 | 83 |
| 4 | CH$_3$CN | Et$_3$N | Cu@MChit/20 | No reaction |
| 5 | MeOH | Cs$_2$CO$_3$ | Cu@MChit/20 | No reaction |
| 6 | MeOH | Na$_2$CO$_3$ | Cu@MChit/20 | 63 |
| 7 | MeOH | K$_2$CO$_3$ | Cu@MChit/20 | 54 |
| 8 | MeOH | KOH | Cu@MChit/20 | 35 |
| 9 | MeOH | Et$_3$N | CuI$_2$/20 | No reaction |
| 10 | MeOH | Et$_3$N | CuI/20 | No reaction |
| 11 | MeOH | Et$_3$N | CuCl/20 | 83 |
| 12 | MeOH | Et$_3$N | CuCl$_2$/20 | No reaction |
| 13 | MeOH | Et$_3$N | CuBr/20 | No reaction |
| 14 | MeOH | Et$_3$N | CuOAc/20 | 78 |
| 15 | MeOH | Et$_3$N | No catalyst | No reaction |
| 16 | MeOH | Et$_3$N | Cu@MChit/15 | 55 |
| 17 | MeOH | Et$_3$N | Cu@MChit/30 | 83 |
| 18 | EtOH | Et$_3$N | Cu@MChit/20 | 75 |

[a] The reaction conditions: 3-allyl-2-thioxo-2,3-dihydroquinazolin-4(1*H*)-one (1 mmol), phenylboronic acid (1 mmol), base (1.2 mmol), solvent (3 mL); [b] isolated yields of the products.

Based on the results in Table 1, it can be observed that the best conditions for the reaction are performing the reaction in methanol as the solvent in the presence of 20 mg of Cu@MChit catalyst using 1.2 equivalent of triethyl amine as the base. The mentioned conditions were selected as the optimized conditions and selected for evaluating the scope and generality of the method. Various 3-substituted 2-thioxo-2,3-dihydroquinazolin-4(1*H*)-

one derivatives were used as the substrates to react with different phenyl boronic acid derivatives bearing a wide range of functionalities. The structure of the synthesized 3-alkyl-2-arylthio-2,3-dihydroquinazolin-4(1*H*)-one derivatives are presented in Table 2. It can be observed that all the substrates have given the desired products in very good isolated yields. It can be observed various 2-arylthio-2,3-dihydroquinazolin-4(1*H*)-ones with benzyl, allyl, aryl, or other alkyl groups have successfully participated in the reaction. In addition, several phenyl boronic acids with different functionalities have been used as substrates and in all cases, the products have been obtained in very good yields.

**Table 2.** Scope and generality of the synthesis of 3-alkyl-2-arylthio-2,3-dihydroquinazolin-4(1*H*)-one [a].

| No. | R$^1$ | R$^2$ | Product | Yield % [b] |
|-----|-------|-------|---------|-------------|
| **4a** | Allyl | 4-Cl-phenyl | | 71 |
| **4b** | Allyl | 4-Br-phenyl | | 80 |
| **4c** | Allyl | 4-Me-phenyl | | 68 |
| **4d** | *i*-propyl | Phenyl | | 76 |
| **4e** | Benzyl | Phenyl | | 82 |
| **4f** | Allyl | 2-Br-phenyl | | 78 |

**Table 2.** *Cont.*

| No. | R$^1$ | R$^2$ | Product | Yield % [b] |
|-----|-------|-------|---------|-------------|
| **4g** | Phenyl | 2-Me-phenyl | | 69 |
| **4h** | Phenyl | Phenyl | | 70 |
| **4i** | Benzyl | 4-Cl-phenyl | | 85 |
| **4j** | 2-phenylethyl | Phenyl | | 72 |
| **4k** | Benzyl | Phenyl | | 65 |
| **4l** | Allyl | Phenyl | | 83 |
| **4m** | Benzyl | 4-Br-phenyl | | 62 |

[a] The reaction conditions: compound **3** (1 mmol, phenylboronic acid (1 mmol), Et$_3$N (1.2 mmol), methanol (3 mL) Cu@MChit (20 mg); [b] isolated yields of the products.

As an interesting property of Cu@MChit, the reusability of the catalyst was studied. To accomplish this, after the completion of the reaction of 3-allyl-2-thioxo-2,3-dihydroquinazolin-4(1*H*)-one with phenylboronic acid, the catalyst was isolated from the reaction mixture and reused in the reaction of the same reactants under the optimized reaction conditions. Cu@MChit was used in five sequential reactions and based on the results, which are presented in Figure 2, the catalyst did not use its activity. The advantageous results of the reusability of the Cu@MChit catalyst confirm its versatility for the synthesis of the desired compounds.

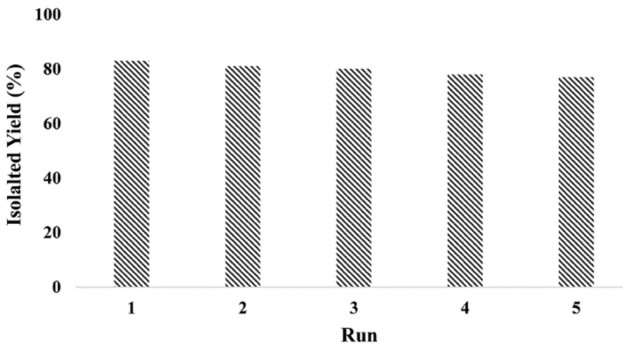

**Figure 2.** The reusability results of Cu@MChit catalyst.

### 3. Materials and Methods

#### 3.1. General Remarks

All the chemicals, reagents, and solvents were purchased from Sigma (Berlin, Germany) and Merck (Berlin, Germany) and were used as received. The reaction performance was monitored by thin-layer chromatography (TLC) on silica gel 254 analytical sheets which were purchased from Fluka (Buchs, Switzerland). Nuclear magnetic resonance (NMR) spectra were recorded on Bruker FT-400 spectrometers (Billerica, IL, USA) using tetramethyl silane (TMS) as the internal standard in pure deuterated solvents. Chemical shifts are given in the δ scale in parts per million (ppm) and singlet (s), doublet (d), triplet (t), multiplet (m), and doublets of doublet (dd) are recorded. The IR spectra were obtained on a Nicolet Magna FT-IR 550 spectrophotometer (Thermo Fisher Scientific, Waltham, MA, USA, potassium bromide disks). Purification of all products was conducted by recrystallization from ethanol.

#### 3.2. Synthesis of Cu@MChit Catalyst

Superparamagnetic iron-oxide-encapsulated silica nanoparticles were synthesized based on our previous report [18]. For this purpose, $Fe^{2+}$ and $Fe^{3+}$ were coprecipitated in a basic media, followed by encapsulation by silica to increase the stability and the active hydroxyl groups on their surface. To a round bottom flask containing 25 mL of thionyl chloride was added superparamagnetic iron oxide coated by silica (1.0 g), and the mixture was sonicated for 30 min and then refluxed and cooled to room temperature. The product was washed with chloroform (25 mL) and sonicated for 30 min. Then, chitosan (1.0 g) and triethylamine (0.5 mL) were added to the reaction mixture and stirred under reflux conditions for 24 h. magnetic chitosan was separated from the reaction mixture and washed with chloroform (3 × 10 mL) and water (3 × 10 mL) and dried in a vacuum oven overnight.

Separately, *N*-phenylacetamide (5 mmol) and $POCl_3$ (5 mmol) were dissolved in DMF and stirred at 80 °C for 12 h. The reaction mixture was poured into the mixture of ice and water and 2-chloroquinoline-3-carbaldehyde was obtained as a precipitate and the product was purified by recrystallization from ethanol. 2-Chloroquinoline-3-carbaldehyde was added to DMF and $Na_2S$ (5 mmol) was added and stirred overnight at 80 °C. Water was added to the reaction mixture and 2-mercaptoquinoline-3-carbaldehyde was extracted by chloroform, and after drying over sodium sulfate, the solvent was evaporated to give the product. The reaction of 2-mercaptoquinoline-3-carbaldehyde with 3,4-diaminobenzoic acid (5 mmol) in DMF (10 mL) in the presence of *p*-toluenesulfonic acid (7 mmol) and stirred at room temperature for 24 h. After the reaction completion, water (10 mL) was

added and the product was extracted by dichloromethane. The product was purified by recrystallization from ethanol.

For the synthesis of the catalyst support, 2-(2-mercaptoquinolin-3-yl)-1*H*-benzo[*d*]imidazole-6-carboxylic acid was activated with 1-ethyl-3-(-3-dimethylaminopropyl) carbodiimide hydrochloride and *N*-hydroxysuccinimide in dichloromethane at room temperature. Then, magnetic starch (1.0 g) was added and stirred for 24 h at room temperature. The product was separated from the reaction mixture by an external magnet and washed with dichloromethane (3 × 10 mL). The product was added to dichloromethane (10 mL) and sonicated for 30 min. Then, CuCl (10 mmol) was added and stirred for 24 h at room temperature. Cu@MChit catalyst was separated by an external magnet and washed with acetone and dried in a vacuum oven for 24 h.

### 3.3. General Procedure for the Synthesis of 2-Amino-N-alkylbenzamide (2)

Isatoic anhydride (815 mg, 5 mmol) and amine (5 mmol) were added to a flask containing water (50 mmol) and stirred for 12 h at room temperature. After the reaction completion, the product was formed as a solid precipitate. The products were separated from the reaction mixture by filtration and washed with water and cold methanol (5 mL). The products were dried overnight at room temperature.

### 3.4. General Procedure for the Synthesis of 3-Substituted 2-thioxo-2,3-dihydroquinazolin-4(1H)-one (3)

To a flask containing ethanol (50 mL) was added compound **2** (5 mmol) and KOH (337 mg, 6 mmol). The reaction was stirred at room temperature and carbon disulfide (380 mg, 5 mmol) was added to the reaction mixture. The mixture was stirred at room temperature for 12 h and then added to ice water. The precipitate was filtered off and purified by recrystallization from ethanol.

### 3.5. General procedure for the Synthesis of 2-Arylthio-2,3-dihydroquinazolin-4(1H)-one Derivatives (4)

Compound **3** (1 mmol) and the derivatives of phenylboronic acid (1 mmol) were added to a flask containing methanol (3 mL) triethyl amine (1.2 mmol) and Cu@MChit (20 mol%). The reaction mixture was stirred under reflux conditions for 12 h. After the reaction was completed, the catalyst was separated from the reaction mixture and the solvent was evaporated and the product was purified by recrystallization from ethanol.

### 3.6. Spectral Data of the Products

The spectral data and a copy of the NMR spectra of compounds **4a–4m** are available in the Supplementary Materials.

3-allyl-2-((4-chlorophenyl)thio)quinazolin-4(*3H*)-one (**4a**)

White solid; m.p. 157–159 °C; $^1$H NMR (400 MHz, DMSO-$d_6$) δ 7.70 (dd, *J* = 7.8, 1.6 Hz, 1H), 7.39–7.25 (m, 4H), 7.16 (d, *J* = 8.8 Hz, 2H), 6.88 (td, *J* = 8.0, 1.3 Hz, 1H), 5.89 (ddt, *J* = 17.2, 10.4, 5.3 Hz, 1H), 5.18 (dd, *J* = 17.2, 1.6 Hz, 1H), 5.09 (dd, *J* = 10.3, 1.7 Hz, 1H), 3.94–3.81 (m, 2H); $^{13}$C NMR (101 MHz, DMSO) δ 168.42, 143.51, 140.70, 135.13, 131.86, 129.34, 129.14, 128.82, 124.73, 120.47, 119.48, 118.78, 115.53, 115.25, 41.30; MS (70 eV): *m*/*z* = 328 (M⁺).

3-allyl-2-((4-bromophenyl)thio)quinazolin-4(*3H*)-one (**4b**)

White solid; m.p. 136–139 °C; $^1$H NMR (400 MHz, DMSO-$d_6$) δ 7.69 (dd, *J* = 7.8, 1.5 Hz, 1H), 7.43 (d, *J* = 8.7 Hz, 2H), 7.37–7.33 (m, 1H), 7.30 (dd, *J* = 8.4, 1.3 Hz, 1H), 7.11 (d, *J* = 8.8 Hz, 2H), 6.95–6.80 (m, 1H), 5.88 (ddt, *J* = 17.2, 10.3, 5.3 Hz, 1H), 5.18 (dd, *J* = 17.2, 1.8 Hz, 1H), 5.09 (dd, *J* = 10.2, 1.6 Hz, 1H), 3.98–3.81 (m, 2H); $^{13}$C NMR (101 MHz, DMSO) δ 168.40, 143.31, 141.17, 135.12, 132.29, 132.00, 131.84, 128.83, 120.73, 119.68, 118.91, 115.70, 115.26, 112.38, 41.30; MS (70 eV): *m*/*z* = 371 (M⁺).

3-allyl-2-(*p*-tolylthio)quinazolin-4(*3H*)-one (**4c**)

White solid; m.p. 166–168 °C; $^1$H NMR (400 MHz, DMSO-$d_6$) δ 7.69 (dd, *J* = 8.0, 1.6 Hz, 1H), 7.29 (ddd, *J* = 8.5, 7.1, 1.5 Hz, 1H), 7.20 (dd, *J* = 8.5, 1.2 Hz, 1H), 7.12 (d, *J* = 8.5 Hz, 2H), 7.05 (d, *J* = 8.4 Hz, 2H), 6.78 (td, *J* = 7.7, 1.2 Hz, 1H), 5.90 (ddt, *J* = 17.3, 10.4, 5.2 Hz, 1H), 5.19 (dd, *J* = 17.2, 1.8 Hz, 1H), 5.09 (dd, *J* = 10.3, 1.7 Hz, 1H), 3.90 (ddd, *J* = 5.6, 3.7, 1.7 Hz, 2H), 2.26 (s, 3H); $^{13}$C NMR (101 MHz, DMSO) δ 168.69, 145.10, 138.64, 135.24, 131.89,

131.08, 129.82, 129.78, 128.71, 120.29, 117.68, 117.37, 115.18, 114.16, 41.28, 20.34; MS (70 eV): $m/z$ = 308 (M$^+$).

3-isopropyl-2-(phenylthio)quinazolin-4(*3H*)-one (**4d**)

White solid; m.p. 161–161 °C; $^1$H NMR (400 MHz, DMSO-$d_6$) δ 8.03 (dd, *J* = 8.0, 1.6 Hz, 1H), 7.69–7.61 (m, 3H), 7.57–7.49 (m, 3H), 7.43–7.37 (m, 1H), 7.11 (dd, *J* = 8.3, 1.1 Hz, 1H), 4.80 (s, 1H), 1.65 (d, *J* = 6.6 Hz, 6H); '$^{13}$C NMR (101 MHz, DMSO) δ 161.10, 156.27, 146.14, 135.76, 134.40, 129.88, 129.32, 127.96, 126.06, 126.00, 125.69, 120.12, 52.64, 19.22; MS (70 eV): $m/z$ = 296 (M$^+$).

**3-benzyl-2-(phenylthio)quinazolin-4(*3H*)-one (4e)**

White solid; m.p. 194–197 °C; $^1$H NMR (400 MHz, DMSO-$d_6$) δ 8.11 (dd, *J* = 8.0, 1.5 Hz, 1H), 7.72 (ddd, *J* = 8.6, 7.2, 1.6 Hz, 1H), 7.60 (dd, *J* = 7.8, 1.8 Hz, 2H), 7.54–7.43 (m, 4H), 7.41–7.28 (m, 5H), 7.23–7.18 (m, 1H), 5.47 (s, 2H); $^{13}$C NMR (101 MHz, DMSO) δ 161.00, 156.65, 146.73, 135.74, 135.52, 134.82, 129.93, 129.35, 128.68, 127.45, 126.72, 126.57, 126.30, 126.11, 118.91, 47.15; MS (70 eV): $m/z$ = 344 (M$^+$).

3-allyl-2-((2-bromophenyl)thio)quinazolin-4(*3H*)-one (**4f**)

White solid; m.p. 144–146 °C; $^1$H NMR (400 MHz, DMSO-$d_6$) δ 8.08 (dd, *J* = 8.0, 1.5 Hz, 1H), 7.86 (dd, *J* = 7.6, 1.7 Hz, 1H), 7.81 (dd, *J* = 7.4, 1.9 Hz, 1H), 7.72–7.66 (m, 1H), 7.55–7.41 (m, 3H), 7.16 (dd, *J* = 8.2, 1.0 Hz, 1H), 6.04 (ddt, *J* = 17.3, 10.2, 5.0 Hz, 1H), 5.35–5.15 (m, 2H), 4.86 (dd, *J* = 4.2, 2.5 Hz, 2H); $^{13}$C NMR (101 MHz, DMSO) δ 160.39, 155.02, 146.72, 138.05, 134.75, 133.58, 132.09, 131.42, 130.06, 129.02, 128.67, 126.47, 126.36, 126.16, 118.89, 117.50, 46.27; MS (70 eV): $m/z$ = 373 (M$^+$).

3-phenyl-2-(*o*-tolylthio)quinazolin-4(*3H*)-one (**4g**)

White solid; m.p. 151–153 °C; $^1$H NMR (400 MHz, DMSO-$d_6$) δ 8.07 (dd, *J* = 7.9, 1.5 Hz, 1H), 7.72 (ddd, *J* = 8.6, 7.1, 1.6 Hz, 1H), 7.66–7.57 (m, 5H), 7.50 (dd, *J* = 7.6, 1.2 Hz, 1H), 7.47–7.38 (m, 3H), 7.28 (td, *J* = 7.1, 2.4 Hz, 1H), 7.21 (dd, *J* = 8.2, 1.1 Hz, 1H), 2.34 (s, 3H); $^{13}$C NMR (101 MHz, DMSO) δ 160.86, 156.20, 147.20, 142.80, 136.53, 136.25, 134.78, 130.64, 130.48, 129.92, 129.56, 129.40, 127.54, 126.70, 126.50, 126.12, 126.04, 119.66, 20.40; MS (70 eV): $m/z$ = 344 (M$^+$).

3-phenyl-2-(phenylthio)quinazolin-4(*3H*)-one (**4h**)

White solid; m.p. 149–152 °C; $^1$H NMR (400 MHz, DMSO-$d_6$) δ 8.08 (dd, *J* = 7.8, 1.6 Hz, 1H), 7.74 (ddd, *J* = 8.5, 7.2, 1.6 Hz, 1H), 7.65–7.55 (m, 7H), 7.52–7.43 (m, 4H), 7.25 (dd, *J* = 8.1, 1.2 Hz, 1H); $^{13}$C NMR (101 MHz, DMSO) δ 160.83, 156.87, 147.13, 136.02, 135.38, 134.80, 129.96, 129.73, 129.52, 129.20, 128.18, 126.51, 126.12, 119.70; MS (70 eV): $m/z$ = 330 (M$^+$).

3-benzyl-2-((4-chlorophenyl)thio)quinazolin-4(*3H*)-one (**4i**)

White solid; m.p. 172–174 °C; $^1$H NMR (400 MHz, DMSO-$d_6$) δ 8.12 (dd, *J* = 8.1, 1.6 Hz, 1H), 7.73 (td, *J* = 8.6, 1.6 Hz, 1H), 7.63 (d, *J* = 8.6 Hz, 2H), 7.57 (d, *J* = 8.6 Hz, 2H), 7.48 (td, *J* = 8.2, 1.1 Hz, 1H), 7.40–7.30 (m, 5H), 7.27–7.24 (m, 1H), 5.45 (s, 2H); $^{13}$C NMR (101 MHz, DMSO) δ 160.97, 156.26, 146.67, 137.28, 135.64, 134.96, 134.86, 129.37, 128.69, 127.48, 126.73, 126.58, 126.51, 126.41, 126.18, 118.94, 47.18; MS (70 eV): $m/z$ = 378 (M$^+$).

3-phenethyl-2-(phenylthio)quinazolin-4(*3H*)-one (**4j**)

White solid; m.p. 188–189 °C; $^1$H NMR (400 MHz, DMSO-$d_6$) δ 8.09 (dd, *J* = 8.0, 1.6 Hz, 1H), 7.70 (ddd, *J* = 8.4, 7.2, 1.7 Hz, 1H), 7.66–7.60 (m, 2H), 7.57–7.51 (m, 3H), 7.45 (t, *J* = 7.5 Hz, 1H), 7.40–7.26 (m, 5H), 7.18 (d, *J* = 8.1 Hz, 1H), 4.36 (t, *J* = 8.0 Hz, 2H), 3.11 (t, *J* = 8.0 Hz, 2H); $^{13}$C NMR (101 MHz, DMSO) δ 160.54, 156.11, 146.64, 137.79, 135.66, 134.62, 129.95, 129.35, 128.79, 128.63, 127.30, 126.70, 126.36, 126.16, 126.04, 118.97, 45.84, 33.53; MS (70 eV): $m/z$ = 358 (M$^+$).

3-(4-fluorobenzyl)-2-(phenylthio)quinazolin-4(*3H*)-one (**4k**)

White solid; m.p. 165–167 °C; $^1$H NMR (400 MHz, DMSO-$d_6$) δ 8.11 (dd, *J* = 8.0, 1.5 Hz, 1H), 7.72 (ddd, *J* = 8.4, 7.1, 1.6 Hz, 1H), 7.61 (dd, *J* = 7.8, 1.9 Hz, 2H), 7.56–7.38 (m, 6H), 7.27–7.16 (m, 3H), 5.44 (s, 2H); $^{13}$C NMR (101 MHz, DMSO) δ 162.66 (d, $^1J_{CF}$ = 242 Hz), 161.02, 156.48, 146.71, 135.53, 134.84, 131.97 (d, $^4J_{CF}$= 3 Hz), 129.95, 129.36, 129.11 (d, $^3J_{CF}$ = 8 Hz), 127.36, 126.56, 126.31, 126.11, 118.93, 115.58 (d, $^2J_{CF}$ = 21 Hz), 46.54; MS (70 eV): $m/z$ = 362 (M$^+$).

3-allyl-2-(phenylthio)quinazolin-4(3*H*)-one (**4l**)

White solid; m.p. 146–148 °C; $^1$H NMR (400 MHz, DMSO-$d_6$) δ 8.08 (dd, $J$ = 8.0, 1.6 Hz, 1H), 7.70 (ddd, $J$ = 8.4, 7.1, 1.6 Hz, 1H), 7.67–7.61 (m, 2H), 7.56–7.49 (m, 3H), 7.44 (td, $J$ = 7.6, 7.1, 1.2 Hz, 1H), 7.19 (dd, $J$ = 8.3, 1.0 Hz, 1H), 6.03 (ddt, $J$ = 17.2, 10.2, 5.0 Hz, 1H), 5.28 (dd, $J$ = 10.5, 1.4 Hz, 1H), 5.21 (dd, $J$ = 17.2, 1.4 Hz, 1H), 4.87–4.75 (m, 2H); $^{13}$C NMR (101 MHz, DMSO) δ 160.44, 156.40, 146.70, 135.56, 134.67, 131.61, 129.91, 129.36, 127.48, 126.44, 126.18, 126.05, 118.89, 117.41, 46.07; MS (70 eV): $m/z$ = 294 (M$^+$).

3-benzyl-2-((4-bromophenyl)thio)quinazolin-4(3*H*)-one (**4m**)

White solid; m.p. 154–156 °C; $^1$H NMR (400 MHz, DMSO-$d_6$) δ 8.12 (dd, $J$ = 8.0, 1.6 Hz, 1H), 7.77–7.67 (m, 3H), 7.55 (d, $J$ = 8.4 Hz, 2H), 7.50–7.45 (m, 1H), 7.42–7.29 (m, 5H), 7.26 (dd, $J$ = 8.2, 1.0 Hz, 1H), 5.45 (s, 2H); $^{13}$C NMR (101 MHz, DMSO) δ 160.97, 156.13, 146.66, 137.44, 135.64, 134.86, 132.29, 128.69, 127.48, 127.03, 126.72, 126.57, 126.41, 126.19, 123.71, 118.94, 47.19; MS (70 eV): $m/z$ = 422 (M$^+$).

## 4. Conclusions

In conclusion, in this paper, a novel catalyst is introduced based on the immobilization of copper onto magnetic chitosan. Chitosan is a green biopolymer from shrimp shells, and as an advantage, this catalyst can be obtained from a polymer with a biological source. Using Cu@MChit, an efficient methodology is introduced for the synthesis of novel 3-alkyl-2-arylthio-2,3-dihydroquinazolin-4(1*H*)-one derivatives. The method is based on a three step reaction from isatoic anhydride, an amine, carbon disulfide, and phenylboronic acid. The method is efficient and the products are obtained in very good isolated yields. The facility of the reactions, the availability of the starting materials, and good yields of the products are the advantages of this method. As an advantage, the catalyst is heterogeneous and simply separable from the reaction mixture using an external magnet. This advantage shortens the reaction workup and removes tedious workup steps for the purification of the products.

**Supplementary Materials:** The following supporting information can be downloaded at: https://www.mdpi.com/article/10.3390/inorganics10120231/s1, spectral data and the copy of the NMR of the products.

**Author Contributions:** Methodology, N.G., A.Y., S.B. and A.M.; formal analysis, S.H., A.I, B.L. and S.M.; writing—original draft preparation, S.B. and M.M.; writing—review and editing, S.B. and M.M.; project administration, M.M. All authors have read and agreed to the published version of the manuscript.

**Funding:** This research received no external funding.

**Conflicts of Interest:** The authors declare no conflict of interest.

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
