# Peer review of "Copper Catalyst-Supported Modified Magnetic Chitosan for the Synthesis of Novel 2-Arylthio-2,3-dihydroquinazolin-4(1H)-one Derivatives via Chan–Lam Coupling"

_inorganics, doi:10.3390/inorganics10120231_

Round 1

Reviewer 1 Report

 M. Mahdavi and co-workers reported “Copper Catalyst Supported Modified Magnetic Chitosan for 2 the Synthesis of Novel 2-Arylthio-2,3-dihydroquinazolin-4(1H)- 3 one Derivatives via Chan-Lam Coupling” methodology.

1. In this report author must mention the importance of Cu@MChit catalyst and reuse of it

2. Even though the reaction is working well with commercially available catalysts such as CuCl then why he chooses Cu@MChit?

3. The catalyst used in the reaction should be mentioned in the mol% terms instead of mg

4. the reaction working with MeOH then why author did not try with other alcoholic solvents such as EtOH etc.,

5. Which substrates author consider for optimization? 

Author Response

  1. Mahdavi and co-workers reported “Copper Catalyst Supported Modified Magnetic Chitosan for 2 the Synthesis of Novel 2-Arylthio-2,3-dihydroquinazolin-4(1H)- 3 one Derivatives via Chan-Lam Coupling” methodology.
  • Thank you very much for the evaluation of our manuscript and for your comments and suggestions.
  1. In this report author must mention the importance of Cu@MChit catalyst and reuse of it
  • Thank you very much for your comment. It is correct and the catalyst could be separated from the reaction mixture using an external magnet.
  1. Even though the reaction is working well with commercially available catalysts such as CuCl then why he chooses Cu@MChit?
  • Thank you very much for your comment. It was mentioned in the conclusion section that: “As an advantage, the catalyst is heterogeneous and simply separable from the reaction mixture using an external magnet. This advantage shortens the reaction workup and removes tedious workup steps for the purification of the products.” It was discussed in the manuscript and highlighted for your kind reference. In addition, it should be noted that 20 mg of Cu@MChit contains much less amount of copper than CuCl (Cu@MChit catalyst contains about 5 mg copper).
  1. The catalyst used in the reaction should be mentioned in the mol% terms instead of mg
  • Thank you very much for your comment. The mol% of the catalyst was calculated and added to the Materials and Methods section, under the synthesis of 2-arylthio-2,3-dihydroquinazolin-4(1H)-one derivative (4) subsection.
  1. the reaction working with MeOH then why author did not try with other alcoholic solvents such as EtOH etc.,
  • The reaction was performed in ethanol according to the reviewer’s suggestion. The reaction was performed and the product was obtained. However, the isolated yield of the product was less than methanol. The result was added to Table 1, entry 18.
  1. Which substrates author consider for optimization? 
  • The reaction conditions, including the substrates that were used to optimize the reaction conditions, were added to the caption of Table 1. The conditions and the substrates are as the following: The reaction conditions: 3-allyl-2-thioxo-2,3-dihydroquinazolin-4(1H)-one (1 mmol), phenylboronic acid (1 mmol), base (1.2 mmol), solvent (3 mL).

Reviewer 2 Report

The manuscript of Mahdavi et al. describes novel efficient catalytic system for the Chan-Lam coupling reaction based on copper immobilized onto magnetic chitosan. The developed catalytic system has been succesively used for the synthesis of 13 examples of quinazolinone derivatives. I consider that the manuscript can be published in Inorganics after response on following minor remark: 1)     During the catalyst screening (Table 1) authors showed that the simple CuCl has the same efficiency as the immobilized catalyst Cu@MChit (compare entries 3 and 11). Authors should highlight advantages of chitosan system or explaine – why they used Cu@MChit instead of CuCl for the further scope inverstigation. 2)     What is known about leaching of copper in course of catalysis? It would be interesting to evaluate the efficiency of a once-used Cu@MChit catalyst when reused in a new reaction. 3)     Authors should recheck data of MS spectroscopy. For example, the calculated value for compound 4h ([M]+) is 330, but the found mass is 332. The same large errors are also observed for compounds 4e and 4g. Details of the device and ionization method that were used to record the MS spectra should be added to the experimental section. Data of HRMS or elemental analysis are desirable for reliable confirmation of purity of organic compounds. 4)     Some typos should be corrected. For example, “isatoc anhydride” in line 44 and “magnetic starch (a.0 g)” in line 165.

Author Response

The manuscript of Mahdavi et al. describes novel efficient catalytic system for the Chan-Lam coupling reaction based on copper immobilized onto magnetic chitosan. The developed catalytic system has been succesively used for the synthesis of 13 examples of quinazolinone derivatives. I consider that the manuscript can be published in Inorganics after response on following minor remark:

  • Thank you very much for the evaluation of our manuscript and your comments and suggestions.

1)     During the catalyst screening (Table 1) authors showed that the simple CuCl has the same efficiency as the immobilized catalyst Cu@MChit (compare entries 3 and 11). Authors should highlight advantages of chitosan system or explaine – why they used Cu@MChit instead of CuCl for the further scope inverstigation. 

  • Thank you very much for your comment. It was mentioned in the conclusion section that: “As an advantage, the catalyst is heterogeneous and simply separable from the reaction mixture using an external magnet. This advantage shortens the reaction workup and removes tedious workup steps for the purification of the products.” It was discussed in the manuscript and highlighted for your kind reference. In addition, it should be noted that 20 mg of Cu@MChit contains much less amount of copper than CuCl (Cu@MChit catalyst contains about 5 mg copper).

2)     What is known about leaching of copper in course of catalysis? It would be interesting to evaluate the efficiency of a once-used Cu@MChit catalyst when reused in a new reaction. 

  • The leaching of copper from Cu@MChit catalyst was studied by stirring the catalyst under the reaction conditions and after 12h, the catalyst was separated from the reaction mixture using an external magnet and the solution was characterized by ICP. No copper was detected in the solution, which confirms the stability of the structure of the catalyst. The results were added to the manuscript.

3)     Authors should recheck data of MS spectroscopy. For example, the calculated value for compound 4h ([M]+) is 330, but the found mass is 332. The same large errors are also observed for compounds 4e and 4g. Details of the device and ionization method that were used to record the MS spectra should be added to the experimental section. Data of HRMS or elemental analysis are desirable for reliable confirmation of purity of organic compounds. 

  • Thank you very much for your comment. The typo was corrected. There was a mistake in the report of the analysis results. Unfortunately, we do not have access to HRMS. Therefore, MS results were added to the manuscript.

4)     Some typos should be corrected. For example, “isatoc anhydride” in line 44 and “magnetic starch (a.0 g)” in line 165.

  • Thank you very much for your careful evaluation of the manuscript. The typos were corrected in the manuscript.

Reviewer 3 Report

The manuscript “Copper Catalyst Supported Modified Magnetic Chitosan for  the Synthesis of Novel 2-Arylthio-2,3-dihydroquinazolin-4(1H)- one Derivatives via Chan-Lam Coupling “ by N. Ghasemi, A. Yavari, S. Bahadorikhalili, A. Moazzam, S. Hosseini, B. Larijani, A. Iraji, S. Moradi, and M. Mahdavi describes the synthesis of the title compounds by a chitosan-based catalyst prepared by the authors.

The manuscript is well written, needing only minor language revision. The approach to these compounds is not new, and in fact it has been a subject of research of the Mahdavi  group for a while, due to the importance of the compounds in question, which may exhit a variety of biological activities when suitably derivatized.

The work seems well substantiated. However I feel that it needs a few small experiments to show the usefulness of the chitosan-supported material as a catalyst.

In this manuscript the authors present in Table 1 the results of the optimization reactions they performed to find the best conditions to carry out the final reaction in the synthesis, the Chan-Lam coupling in which the new catalyst is used. However, they found that the reaction catalyzed by the chitosan-based material (entry 3) gives the same result as a reaction performed with CuCl as catalyst (entry 11) under the same reaction conditions (solvent: MeOH, base: Et3N, yield 83%).

From the results it appears that the new catalyst brings no advantage to this synthesis. All you need to do is pick up a bottle of CuCl and perform the reaction, rather than spend a lot of time and reagents preparing a catalyst that will give the same result. Even if the actual amount of copper used is less.

To make this work worthwhile, it would be useful if the authors showed with a few simple reactions that the catalyst may be reutilized (recycled). The last reaction could be performed a few times on a standard substrate with the same catalyst if this could be filtered off, dried, and reused.

I suggest that the authors try this simple reactions, and resubmit the manuscript. No new product characterizations are required, so it should be relatively simple and fast to perform.

Besides this point, there are a few minor corrections to be made:

1)    Table 2: The reaction in which the products are obtained should appear on top of Table 2. Not only the general structure of the final product.

2)     Instead of the amount of catalyst used being shown in mgs, the mol% in relation to the substrate should be indicated.

3)    The references should show journal abbreviations.

Other corrections required:

1)    P2, line 34: “a benzene ring with an amine and a carboxylic acid or cyanide group in ortho positions “ => “a benzene derivative bearing an amine and a carboxylic acid or cyanide group in ortho positions”

2)    P2, line 37:  “We have extended” => “we have developed”; at the second occurrence use “reported” instead of developed”

3)    P2, line 41: =”on extending methods for the synthesis” => “on developing methods for the synthesis”

4)    Line 44: “isatoc anhydride” => isatoic anhydride”

5)    Line 47: “synthesis rout” => “synthetic route”

6)    P4, paragraph 1: the word efficient appears 3 times. Please modify.

7)    P4, line 86: “The significant point of this step” => “The driving force for this step”

8)    Page 7, line 165: magnetic startch (a,0 g) => amount missing

Author Response

The manuscript “Copper Catalyst Supported Modified Magnetic Chitosan for  the Synthesis of Novel 2-Arylthio-2,3-dihydroquinazolin-4(1H)- one Derivatives via Chan-Lam Coupling “ by N. Ghasemi, A. Yavari, S. Bahadorikhalili, A. Moazzam, S. Hosseini, B. Larijani, A. Iraji, S. Moradi, and M. Mahdavi describes the synthesis of the title compounds by a chitosan-based catalyst prepared by the authors. The manuscript is well written, needing only minor language revision. The approach to these compounds is not new, and in fact it has been a subject of research of the Mahdavi  group for a while, due to the importance of the compounds in question, which may exhit a variety of biological activities when suitably derivatized.

The work seems well substantiated. However I feel that it needs a few small experiments to show the usefulness of the chitosan-supported material as a catalyst.

  • Thank you very much for the evaluation of our manuscript and for your comments and suggestions.

In this manuscript the authors present in Table 1 the results of the optimization reactions they performed to find the best conditions to carry out the final reaction in the synthesis, the Chan-Lam coupling in which the new catalyst is used. However, they found that the reaction catalyzed by the chitosan-based material (entry 3) gives the same result as a reaction performed with CuCl as catalyst (entry 11) under the same reaction conditions (solvent: MeOH, base: Et3N, yield 83%).

From the results it appears that the new catalyst brings no advantage to this synthesis. All you need to do is pick up a bottle of CuCl and perform the reaction, rather than spend a lot of time and reagents preparing a catalyst that will give the same result. Even if the actual amount of copper used is less.

  • Thank you very much for your comment. It was mentioned in the conclusion section that: “As an advantage, the catalyst is heterogeneous and simply separable from the reaction mixture using an external magnet. This advantage shortens the reaction workup and removes tedious workup steps for the purification of the products.” It was discussed in the manuscript and highlighted for your kind reference. In addition, it should be noted that 20 mg of Cu@MChit contains much less amount of copper than CuCl (Cu@MChit catalyst contains about 5 mg copper).

To make this work worthwhile, it would be useful if the authors showed with a few simple reactions that the catalyst may be reutilized (recycled). The last reaction could be performed a few times on a standard substrate with the same catalyst if this could be filtered off, dried, and reused.

I suggest that the authors try this simple reactions, and resubmit the manuscript. No new product characterizations are required, so it should be relatively simple and fast to perform.

  • Thank you very much for your comment. The experiments were performed according to your suggestion and the results were added to the manuscript. It was observed that Cu@MChit catalyst was highly active in the reaction after 5 sequential reactions.

Besides this point, there are a few minor corrections to be made:

1)    Table 2: The reaction in which the products are obtained should appear on top of Table 2. Not only the general structure of the final product.

  • The reaction was added to the Table 2.

2)     Instead of the amount of catalyst used being shown in mgs, the mol% in relation to the substrate should be indicated.

  • The mol% of the catalyst was measured and added to the materials and methods in the manuscript.

3)    The references should show journal abbreviations.

  • It was corrected.

Other corrections required:

1)    P2, line 34: “a benzene ring with an amine and a carboxylic acid or cyanide group in ortho positions “ => “a benzene derivative bearing an amine and a carboxylic acid or cyanide group in ortho positions”

2)    P2, line 37:  “We have extended” => “we have developed”; at the second occurrence use “reported” instead of developed”

3)    P2, line 41: =”on extending methods for the synthesis” => “on developing methods for the synthesis”

4)    Line 44: “isatoc anhydride” => isatoic anhydride”

5)    Line 47: “synthesis rout” => “synthetic route”

6)    P4, paragraph 1: the word efficient appears 3 times. Please modify.

7)    P4, line 86: “The significant point of this step” => “The driving force for this step”

8)    Page 7, line 165: magnetic startch (a,0 g) => amount missing

  • Thank you very much for your comment. The typo was corrected.

Round 2

Reviewer 1 Report

The author properly responded to the reviewers' comments. Therefore am happy to accept the current form for publication

Author Response

Dear reviewer,

Thank you very much for your evaluation of our manuscript and for your kind comment.

Reviewer 3 Report

I am satisfied with the corrections made by the authors. However, there is one small error still. In the equation in Table 2, the nitrogen substituent appears as CH2R on the substrate and as R1 in the product. They should be identical. Either CH2R1 on both sides or R1 on both sides,.

That is, is the first nitrogen substituent CH2-allyl or simply allyl?

Please correct before submitting.

Author Response

I am satisfied with the corrections made by the authors. However, there is one small error still. In the equation in Table 2, the nitrogen substituent appears as CH2R on the substrate and as R1 in the product. They should be identical. Either CH2R1 on both sides or R1 on both sides,.

That is, is the first nitrogen substituent CH2-allyl or simply allyl?

Please correct before submitting.

Answer: Dear reviewer. Thank you very much for your careful point. The equation was corrected in the manuscript. The substituent is allyl group, as corrected in the table entry and in the equation.